

# The effect of internet addiction and smartphone addiction on sleep quality among Turkish adolescents

Ayla Acikgoz[1], Burcu Acikgoz[2] and Osman Acikgoz[3]

[1] Vocational School of Health Services, Dokuz Eylül University, Izmir, Turkey
[2] Department of Physiology, Graduate School of Health Sciences, Dokuz Eylül University, Izmir, Turkey
[3] Department of Physiology, Faculty of Medicine, Dokuz Eylül University, Izmir, Turkey

## ABSTRACT

**Background:** Sleep quality plays a principal role in the protection of health. There is an increasing number of studies in the literature demonstrating that internet addiction and smartphone addiction impair sleep quality. However, the number of studies on Turkish adolescents is very limited. Therefore, this study examined the effects of internet addiction and smartphone addiction on sleep quality among Turkish adolescents.

**Methods:** Participants in this cross-sectional study were 910 adolescents aged 13–18 years. Data were collected with the Short Internet Addiction Test, Smartphone Addiction Scale, and Pittsburgh Sleep Quality Index. In addition, a questionnaire was used to gather information about the demographic, socioeconomic, and health-related characteristics. Pearson's Chi-square test, Chi-square test for trend, Mann–Whitney $U$ test, logistic regression analysis, and Spearman's correlation analysis were used in the analysis.

**Results:** The sleep quality of 58.7% of the adolescents was poor. Additionally, girls and adolescents ≥16 years old had poor sleep quality. Sleep quality deteriorated as perceived health status and perceived economic status of family deteriorated. Compared to participants with normal internet addiction scores, poor sleep quality was 1.83 (95% CI [1.22–2.74]) times higher in those with problematic internet addiction and 1.99 (95% CI [1.23–3.87]) times higher in those with pathological internet addiction. One point increase in Smartphone Addiction Scale total score increased poor sleep quality 1.01 (95% CI [1.00–1.02]) times. Sleep quality scale were positively correlated with the smartphone addiction and internet addiction. However, there was no positive correlation between habitual sleep efficiency subcomponent of sleep quality and smartphone addiction and internet addiction.

**Conclusions:** Internet addiction and smartphone addiction were associated with poor sleep quality in adolescents. Older adolescents (≥16 years), gender (female), poor health perception, and perception of moderate economic status of the family were other factors associated with poor sleep quality.

Corresponding author
Ayla Acikgoz,
ayla.acikgoz@deu.edu.tr

## INTRODUCTION

The internet, a new communication, education, and entertainment medium, has become an indispensable part of our daily life of its users recently. The proportion of young people aged between 15 and 24 years old using the internet has reached 98% in developed countries and 66% in developing countries (*International Telecommunication Union Development Sector, 2021*). Excessive internet use has been described in different terms, including internet addiction (IA), pathological internet use, internet use disorder, and problematic internet use. IA is the most used term describing excessive internet use (*Shaw & Black, 2008*). IA, which is defined as the inability of a person to control internet use, may impair an individual's psychological state and lead to problems in daily life (*Pawlikowski, Altstötter-Gleich & Brand, 2013*). IA can have adverse effects on many lifestyle-related factors in adolescents, including time management, irregular eating habits, physical dysfunction, shortened sleep duration, and poor sleep quality (SQ). Moreover, IA is associated with negative health outcomes such as anxiety disorder, stress, depression, attention deficit/hyperactivity disorder, self-identity confusion, low self-esteem, social anxiety, and suicide attempt (*Choi et al., 2015*; *Koças & Şaşmaz, 2018*; *Onat et al., 2019*; *Salama, 2020*; *Lachmann et al., 2017*).

Smartphones are devices that provide access to the internet in any environment. The use of smartphones has been increasing. While 6.4 billion people are reported to be using smartphones in 2021, this number is expected to reach 7.5 billion in 2026 (*Statista, 2021*). Although smartphones have made our lives more convenient due to their numerous benefits such as providing faster and easier communication, being a means for socialization and entertainment, facilitating access to information, and improving time management, excessive use may result in smartphone addiction (SA) (*Kwon et al., 2013*; *Panova & Carbonell, 2018*). There are no official diagnostic criteria for SA. Based on the definition of IA, SA can be defined as the excessive use of smartphones that disrupts the daily life of its users (*Demirci, Akgönül & Akpinar, 2015*). The components of SA are compulsive behavior, tolerance, withdrawal, and functional impairment (*Kwon et al., 2013*; *Panova & Carbonell, 2018*). SA is stated to directly cause a variety of health symptoms. Since SA leads to a sedentary lifestyle in adolescents, it increases the incidence of physical symptoms such as lack of physical energy, physiological dysfunction and weakened immunity (*Wang et al., 2021*; *Xie, Dong & Wang, 2018*). Although it has not been clearly defined yet, SA mostly occurs with symptoms such as not being able to stay away from the phone, checking the phone frequently, insomnia due to excessive smartphone use, and deterioration of SQ (*Wang et al., 2021*; *Kocamaz et al., 2020*; *Kurugodiyavar et al., 2018*).

Sleep is one of the basic life activities that affect the quality of life and health of individuals and is a concept with physiological, psychological and social dimensions. Good sleep is essential for survival (*Brown et al., 2018*). It plays a principal role in the protection of health and the functioning of central nervous, immune, hormonal, and cardiovascular systems (*Brown et al., 2018*; *Kazama, Maruyama & Nakamura, 2015*; *Nam, Han & Lee, 2017*; *Medic, Wille & Hemels, 2017*; *Tarokh, Saletin & Carskadon, 2016*). SQ consists of five dimensions: sleep duration, sleep continuity or efficiency, timing,

alertness/sleepiness, and satisfaction/quality (*Buysse et al., 1989*). Adequate sleep and good SQ are vital in strengthening physical growth and academic performance in adolescents (*Bilgrami et al., 2017*; *Lemma et al., 2014*). Poor SQ is associated with depressive symptoms, poor connection with the school, impaired daytime functioning, weaker immune system, and fatigue (*Bilgrami et al., 2017*; *Brown et al., 2018*). In the literature, it is stated that age, gender, psychosocial, socioeconomic and cultural factors play a role on SQ as well as daily living habits that are at the level of addiction (*Huang et al., 2020*; *Knutson, 2013*; *Kurugodiyavar et al., 2018*; *Wang et al., 2019*; *Özcan, 2020*). These factors can be confounding factors when examining the effect of IA and SA on SQ. In the literature, studies conducted with adolescents indicate that the frequency of poor SQ increases with age (*Demirci et al., 2015*; *Wang et al., 2019*). It is stated that sleep deprivation increases the daytime sleepiness and careless behaviors of adolescents and affects their behavioral problems. Poor SQ is an important problem that needs to be treated during adolescence with emotional, cognitive or behavioral problems (*Demirci, Akgönül & Akpinar, 2015*; *Tarokh, Saletin & Carskadon, 2016*).

Adolescence is defined by World Health Organization as the life stage between childhood and adulthood, between the ages of 10 and 19. Adolescence are grouped into three overlapping age groups: early adolescence (the age group between 10 and 15 years), middle adolescence (the age group between 14 and 17 years), late adolescence (the age group between 16 and 19 years). Because the changes seen in adolescents are not constant and occur at different ages for different adolescents, there is overlap in age groups (*World Health Organization, 2010*). Adolescence is a critical period when a person is prone to addiction (*Brown et al., 2018*). Internet use and smartphones have become common among adolescents and young adults (*Kurugodiyavar et al., 2018*; *Lachmann et al., 2017*; *Lee, Kim & Choi, 2017*). According to the data of 2021 in Turkey, 96.7% of people in the 16–24 age group use smartphones and 95.7% use the internet (*Turkish Statistical Institute, 2021*). Considering that the internet and/or smartphones are used extensively by adolescents (*Bilgrami et al., 2017*; *Choi et al., 2015*; *Lee, Kim & Choi, 2017*), their effects on sleep disorders, which are associated with many health problems, should be examined closely. Many studies have shown an association between excessive internet and/or smartphone use and bedtime procrastination, shorter sleep duration, and sleep disturbances in adolescents (*Bruni et al., 2015*; *de Lima & Silva, 2018*; *Huang et al., 2020*; *Kang et al., 2020*; *Lin et al., 2019*; *Wang et al., 2021*; *Xie, Dong & Wang, 2018*). Moreover, poor SQ in adolescence negatively affects learning, memory, attention, and cognition, therefore reduces academic success (*Brown et al., 2018*; *Tarokh, Saletin & Carskadon, 2016*). Studies in Turkey investigating the effects of IA and/or SA on sleep quality have mostly been conducted on adults or university students (*Demirci et al., 2015*; *Demirci, Akgönül & Akpinar, 2015*; *Kocamaz et al., 2020*; *Özcan, 2020*). However, there are few studies conducted on adolescents (*Akçay & Akçay, 2018*; *Ekinci et al., 2014*; *Eliacik et al., 2016*; *Koças & Şaşmaz, 2018*; *Onat et al., 2019*). There are some limitations in most of these studies examining the relationship between IA and SA, which are important addictions for adolescents of the last quarter century, and sleep quality. Two of these studies were conducted on the patient group (*Eliacik et al., 2016*; *Onat et al., 2019*). In another study, SQ

was determined with a semi-structured scale whose validity and reliability were not determined (*Ekinci et al., 2014*). In one of the two studies conducted with healthy adolescents, the sample size was relatively small and IA and SA were not determined by scales, and the correlation between media (smartphone, computer, television) use and sleep quality was examined (*Akçay & Akçay, 2018*). In a study with a sufficient sample size and in which IA and SQ were determined with appropriate scales, the most important factor causing poor SQ was found to be IA (*Koças & Şaşmaz, 2018*). There is no study in the literature examining the effect of SA on SQ in adolescents in Turkey. Therefore, new studies examining the effects of IA and SA on SQ are needed to compensate for this gap in the literature. Internet usage in the adolescent age group is very high in Turkey (*International Telecommunication Union Development Sector, 2021*). It is stated that IA and SA addiction and the risk of addiction are high in this age group (*Eliacik et al., 2016*; *Koças & Şaşmaz, 2018*; *Onat et al., 2019*). It is important to relate these addictions to potential health consequences that may affect adolescent health. Studies are needed to determine the possible effect on sleep quality in order to detect and prevent IA and SA status among high school students and to make treatment interventions.

Within this scope, this study aimed to investigate the effect of IA and SA on SQ in Turkish adolescents and to identify the sociodemographic factors affecting SQ.

Based on the aforementioned information, the following hypotheses have been developed.

Hypothesis 1: IA negatively affects SQ in adolescents.

Hypothesis 2: SA negatively affects SQ in adolescents.

Hypothesis 3: Adolescents' sociodemographic (age, gender, body mass index, health status) and familial characteristics (parental education and family economic status) negatively affect SQ.

## MATERIALS AND METHODS

This cross-sectional study was conducted with Turkish adolescents aged 13–18 years in Izmir—Turkey between March and April 2018. Participants consisted of students studying at a public high school ($N = 593$) and two private high schools ($N = 659$). Students at the public high school were from different parts of the city as they were enrolled in the school according to the results of a central exam. The students of the two private schools were also from different parts of the city and some were staying in the dormitory of the school. Therefore, the participants had different socio-cultural backgrounds, which may partially reflect the general society. All students were invited to complete the questionnaires of the study, with no sampling. 910 of the 1,252 students completed the questionnaires and the response rate was 72.7%.

All participants were assessed using three standardized questionnaires: The Short Internet Addiction Test (s-IAT) (*Pawlikowski, Altstötter-Gleich & Brand, 2013*), Smartphone Addiction Scale (SAS) (*Kwon et al., 2013*), and Pittsburgh Sleep Quality Index (PSQI) (*Buysse et al., 1989*). In addition, another questionnaire was used to gather the demographic, socioeconomic, and health-related characteristics. Questionnaires were distributed to the students and written informed consent of their families was obtained.

The students filled out the questionnaires in the classroom under the supervision of the researchers.

The socioeconomic and health status levels of the students were determined according to their individual perceptions. In order to determine the perceived economic level, the students were asked the question of "How do you perceive your family's economic situation?" In order to determine the perceived health status, the students were asked the question of "How is your health in general?" The perceived economic and health situation were evaluated with a five-point Likert scale according to the answers given (very good, good, moderate, bad, very bad) (*Bishop & Herron, 2015*). Since the number of students who stated the perception of economic situation as "very good" and "very bad" was very few, this variable was grouped as "very bad-bad" and "very good-good" in the analysis.

The mean age of the students participating in the study was 15.7 ± 1.20. In the literature, studies conducted with adolescents indicate that the frequency of poor SQ increases with age (*Demirci et al., 2015*; *Wang et al., 2019*). Therefore, in our study, adolescents were divided into two approximately equal groups (≤15 and ≥16) to examine the effect of age.

Using the height and weight values declared by each student, the body mass index (BMI) (kg/m$^2$) of the students was calculated. BMI values were classified according to their percentile values (underweight: <5th percentile, healthy weight: 5th to <85th percentile, overweight: 85th to <95th percentile, obese: ≥95th percentile) (*World Health Organization, 2021*).

### Internet addiction (IA)

Internet addiction was measured using the s-IAT (*Pawlikowski, Altstötter-Gleich & Brand, 2013*). Cronbach's alpha coefficient for the original version of the s-IAT is 0.85 (*Pawlikowski, Altstötter-Gleich & Brand, 2013*). It is a five-point Likert scale (1 = never, 5 = very often) (*Bishop & Herron, 2015*) consisting of 12 items. The total score is between 12 and 60. If the score obtained in the questionnaire is below 31, it is classified as having normal internet use, if it is between 31 and 37, it shows problematic internet use, and above 37, it is defined as pathological internet use (*Pawlikowski, Altstötter-Gleich & Brand, 2013*). The Turkish version of the s-IAT is a reliable and valid scale for adolescents, and its Cronbach's alpha coefficient is 0.86 (*Kutlu et al., 2016*). In our study, the Cronbach's alpha coefficient was 0.78.

### Smartphone addiction (SA)

The SAS was used to evaluate SA (*Kwon et al., 2013*). Cronbach's alpha coefficient for the original version of the SAS is 0.96 (*Kwon et al., 2013*). It contains six subscales of SA: daily-life disturbance, positive anticipation, withdrawal, cyberspace-oriented relationship, overuse, and tolerance. "Daily-life disturbance" includes inability to work planned, difficulty in concentrating, drowsiness or blurred vision, pain on the wrists or neck, and sleeping disturbance. "Positive anticipation" is defined as getting excited and relieved of stress with smartphone use and feeling empty without a smartphone. "Withdrawal" includes being impatient, moody, and unbearable without a smartphone, constantly thinking about using a smartphone, never giving up on using the smartphone, and getting

angry when disturbed while using the smartphone. "Cyberspace-oriented relationship" is defined as the feeling that one's relationships with friends acquired through smartphones are more intimate than relationships with real-life friends, and an uncontrolled sense of loss when one cannot use their smartphone. "Overuse" is defined as a person using their smartphone uncontrollably, preferring to solve a problem using the smartphone instead of asking for help from others, and wanting to use their smartphone again immediately after using it. "Tolerance" is defined as a person trying to limit the use of smartphone but failing to do so (*Kwon et al., 2013*). SAS is a six-point Likert scale (1 = strongly disagree, 6 = strongly agree) (*Bishop & Herron, 2015*) consisting of 33 items. The total score is between 33 and 198, a cut-off point was not suggested. The higher scores on the scale indicate that the addiction is more severe (*Kwon et al., 2013*). The Turkish version of the SAS is a reliable and valid and its Cronbach's alpha coefficient is reported to be 0.95 (*Demirci et al., 2014*). In our study, the Cronbach's alpha coefficient was 0.86.

### Sleep quality (SQ)

Sleep quality was measured using the PSQI (*Buysse et al., 1989*). Cronbach's alpha coefficient for the original version of the PSQI is 0.83 (*Buysse et al., 1989*). It consists of seven components: subjective SQ (sleep quality as perceived by the subject), sleep latency (time from going to bed to falling asleep), sleep duration (time spent asleep), habitual sleep efficiency (effective sleeping time of total time spent in bed), sleep disturbances (coughing or snoring loudly, feeling too cold and/or hot, having bad dreams, not breathing comfortably, and waking up in the middle of the night or early morning), use of sleep medication (taking medication to help sleep), and daytime dysfunction (difficulty in staying awake, difficulty in doing tasks during the day) (*Buysse et al., 1989*). The answer to each question is different in PSQI. A four-point Likert scale was used in only two questions (*Bishop & Herron, 2015*). The total score is between 0 and 21. Higher scores indicate poor SQ. The suggested cut-off value was used to distinguish between good sleepers (score <5) and poor sleepers (score ≥5) (*Ağargün, Kara & Anlar, 1996*; *Buysse et al., 1989*). The Turkish version of the PSQI is reliable and valid. Its Cronbach's alpha coefficient is 0.79 (*Ağargün, Kara & Anlar, 1996*). In our study, the Cronbach's alpha coefficient was 0.64.

### Statistical analysis

Sociodemographic and individual characteristics of the adolescents were presented as numbers and percentages. Continuous variables were presented as mean ± standard deviation. We analyzed the normality of the data with the Kolmogorov–Smirnov test. As our data were not normally distributed ($p < 0.05$), we used non-parametric tests. Categorical variables were summarized using percentages and compared using the Chi-square test and Chi-square test for trend. The Mann–Whitney $U$ test was used to compare the SAS subscale scores in the SQ groups. The relationship between SAS total score, SAS subscale scores, sIAT total score, and PSQI components was determined by the Spearman's correlation analysis. The effect of students' sociodemographic and individual characteristics, IA, and SA on poor SQ was analyzed using the adjusted binary logistic

regression (enter method). Statistical analysis was conducted using the IBM Statistical Package of the Social Science for Windows 24.0 (IBM Corp., Armonk, NY, USA) statistics package software. The posthoc power analysis of the study was performed using G*Power software and found to be 0.99 ($n$ = 910, $\alpha$ = 0.05, effect size = 0.50).

## Ethics

The study was conducted in accordance with the principles of the Helsinki Declaration. All students were informed about the aims of the study, and parents signed a written informed consent indicating that their children voluntarily participated in the study. The questionnaires were filled in classrooms. Ethical approval was obtained from the Non-invasive Research Ethics Committee of Dokuz Eylul University, Izmir, Turkey (Decision No: 2018/01-39).

## RESULTS

Nine hundred and ten Turkish adolescents were included in this study. The SQ of 58.7% of the students was poor while 41.3% had a good SQ. The relationship between some sociodemographic and individual characteristics of the students with their SQ is presented in Table 1. As shown in Table 1, gender, age, paternal education level, perceived health status and perceived economic status of family are significantly associated with sleep quality ($p < 0.01$; $p = 0.02$; $p = 0.02$; $p < 0.01$; $p < 0.01$, respectively). The SQ of female students, students in the ≥16 age group, and those whose father's education level ≤ high school were significantly poor. The worse the perceived health status of the students was, the poorer the SQ of the students became. Additionally, as the perceived economic status of family deteriorated, the SQ of the students deteriorated significantly.

The IA scores of 72.4% of the students were "normal," 19.2% were "problematic" and 8.4% were "pathological." As the IA level of the students increased, the SQ also worsened significantly (Table 2).

The SAS total score of the students was found to be 77.9 ± 27.7 (Min: 33.0, Max: 193.0). The relationship between SA and SQ is presented in Table 3. Students with poor SQ had significantly higher SAS total and sub-components scores.

The correlation between SAS and PSQI scores is shown in Table 4. There was a weak positive correlation between the scores of some subcomponents of SAS, SAS total score, the total score of sIAT, and PSQI subcomponent scores (except for the habitual sleep efficiency). There was a weak positive correlation between the daily-life disturbance and overuse subcomponent scores of the SAS and the PSQI subcomponent scores (except habitual sleep efficiency) (daily-life disturbance; $r = 0.241$; $p < 0.01$, $r = 0.120$; $p < 0.01$, $r = 0.158$; $p < 0.01$, $r = 0.274$; $p < 0.01$, $r = 0.123$; $p < 0.01$, $r = 0.382$; $p < 0.01$, overuse; $r = 0.146$; $p < 0.01$, $r = 0.071$; $p = 0.03$, $r = 0.117$; $p < 0.01$, $r = 0.182$; $p < 0.01$, $r = 0.073$; $p = 0.02$, $r = 0.257$; $p < 0.01$, respectively). There was a weak positive correlation between the positive anticipation subcomponent score of the SAS and the PSQI subcomponents sleep disturbances ($r = 0.122$; $p < 0.01$), use of sleep medication ($r = 0.093$; $p < 0.01$) and daytime dysfunction scores ($r = 0.140$; $p < 0.01$). There was a weak positive correlation between SAS withdrawal and cyberspace-oriented relationship subcomponent scores and

**Table 1 Relationship between some sociodemographic and individual characteristics of the students and their sleep quality ($n = 910$).**

| Characteristics | | Sleep quality | | $\chi^2$-value | $p^*$ |
|---|---|---|---|---|---|
| | | Poor ($n = 534$) $n$ (%) | Good ($n = 376$) $n$ (%) | | |
| Age (years) | ≥16 | 301 (62.2) | 183 (37.8) | 5.25 | **0.022** |
| | ≤15 | 233 (54.7) | 193 (45.3) | | |
| Gender | Female | 289 (64.4) | 160 (35.6) | 11.81 | **0.001** |
| | Male | 245 (53.1) | 216 (46.9) | | |
| Paternal education level | ≤High school | 255 (62.8) | 151 (37.2) | 5.15 | **0.023** |
| | University | 279 (55.4) | 225 (44.6) | | |
| Maternal education level | ≤High school | 213 (60.3) | 140 (39.7) | 0.65 | 0.419 |
| | University | 321 (57.6) | 236 (42.4) | | |
| Perceived health status | Very good-good | 426 (55.9) | 336 (44.1) | 18.01 | **<0.001#** |
| | Moderate | 85 (69.7) | 37 (30.3) | | |
| | Bad-very bad | 23 (88.5) | 3 (11.5) | | |
| Perceived economic status of family | Very good-good | 328 (55.0) | 268 (45.0) | 9.49 | **0.003#** |
| | Moderate | 194 (65.5) | 102 (34.5) | | |
| | Bad-very bad | 12 (66.7) | 6 (33.3) | | |
| Body Mass Index | Underweight | 12 (70.6) | 5 (29.4) | 2.86 | 0.414 |
| | Normal | 412 (59.5) | 280 (40.5) | | |
| | Overweight | 87 (53.7) | 75 (46.3) | | |
| | Obese | 23 (59.0) | 16 (41.0) | | |

Notes:
*Pearson's Chi-square test.
#Chi-square test for trend.
Significant $p$ values were shown in bold.

**Table 2 Relationship between internet addiction and sleep quality.**

| Internet addiction | Sleep quality | | | | $\chi^2$-value | $p^*$ |
|---|---|---|---|---|---|---|
| | Poor $n$ | % | Good $n$ | % | | |
| Normal | 346 | 52.5 | 313 | 47.5 | 38.90 | **<0.001** |
| Problematic | 127 | 72.6 | 48 | 27.4 | | |
| Pathological | 61 | 80.3 | 15 | 19.7 | | |

Notes:
*Chi-square test for trend.
Significant $p$ values were shown in bold.

PSQI subcomponent scores (excluding sleep latency and habitual sleep efficiency) (withdrawal; $r = 0.099$; $p < 0.01$, $r = 0.090$; $p < 0.01$, $r = 0.215$; $p < 0.01$, $r = 0.093$; $p < 0.01$, $r = 0.215$; $p < 0.01$, cyberspace-oriented relationship; $r = 0.113$; $p < 0.01$, $r = 0.080$; $p = 0.01$, $r = 0.170$; $p < 0.01$, $r = 0.128$; $p < 0.01$, $r = 0.189$; $p < 0.01$, respectively). There was a weak positive correlation between the tolerance subcomponent score of the SAS and the

**Table 3 Relationship between smartphone addiction and sleep quality.**

| Sleep quality | SAS Subscales | | | | | | SAS total score |
|---|---|---|---|---|---|---|---|
| | Daily-life disturbance | Positive anticipation | Withdrawal | Cyberspace-oriented relationship | Overuse | Tolerance | |
| Good (n = 376) | 9.48 ± 4.32 | 16.36 ± 7.08 | 12.72 ± 5.82 | 14.02 ± 5.26 | 10.39 ± 4.46 | 6.97 ± 3.99 | 69.95 ± 23.74 |
| Poor (n = 534) | 12.65 ± 5.48 | 18.52 ± 8.28 | 15.15 ± 6.92 | 16.12 ± 6.58 | 12.81 ± 5.08 | 8.30 ± 4.24 | 83.55 ± 28.97 |
| $p^*$ | **<0.001** | **<0.001** | **<0.001** | **<0.001** | **<0.001** | **<0.001** | **<0.001** |

Notes:
*Mann–Whitney $U$ test.
Significant $p$ values were shown in bold.

**Table 4 Correlation between the smartphone addiction scale and Pittsburgh Sleep Quality Index scores.**

| SAS subscales | Pittsburgh Sleep Quality Index scores | | | | | | |
|---|---|---|---|---|---|---|---|
| | Subjective sleep quality | Sleep latency | Sleep duration | Habitual sleep efficiency | Sleep disturbances | Use of sleep medication | Daytime dysfunction |
| Daily-life disturbance | **0.241**** | **0.120**** | **0.158**** | 0.033 | **0.274**** | **0.123**** | **0.382**** |
| Positive anticipation | 0.017 | 0.029 | 0.038 | −0.045 | **0.122**** | **0.093*** | **0.140**** |
| Withdrawal | **0.099*** | 0.040 | **0.090*** | −0.054 | **0.215**** | **0.093*** | **0.215**** |
| Cyberspace-oriented relationship | **0.113**** | 0.051 | **0.080*** | 0.007 | **0.170**** | **0.128**** | **0.189**** |
| Overuse | **0.146**** | **0.071*** | **0.117**** | −0.063 | **0.182**** | **0.073*** | **0.257**** |
| Tolerance | **0.108**** | 0.025 | **0.097*** | 0.002 | **0.166**** | 0.062 | **0.237**** |
| SAS total score | **0.149**** | **0.070*** | **0.121**** | −0.030 | **0.238**** | **0.123**** | **0.292**** |
| Internet addiction total score | **0.202**** | **0.129**** | **0.142**** | 0.030 | **0.217**** | **0.086*** | **0.310**** |

Notes:
*$p < 0.05$.
**$p < 0.001$.
The results are expressed as Rho value.
Significant $p$ values were shown in bold.

subjective SQ ($r = 0.108$; $p < 0.01$), sleep duration ($r = 0.097$; $p < 0.01$), sleep disturbances ($r = 0.166$; $p < 0.01$), and daytime dysfunction scores ($r = 0.237$; $p < 0.01$) of the PSQI subcomponents (Table 4).

A logistic regression analysis of variables affecting poor SQ is shown in Table 5. Poor SQ was 1.39 (95% CI [1.05–1.84]; $p = 0.02$) times higher in the ≥16 age group than in the ≤15 age group. Poor SQ was 1.50 (95% CI [1.13–2.00]; $p < 0.01$) times higher in female students than male students. Compared to those who perceived their health status as "very good-good," poor sleep quality was 1.74 (95% CI [1.12–2.69]; $p = 0.01$) times higher in those who perceived their health status as "moderate," and 6.39 (95% CI [1.83–22.30]; $p < 0.01$) times higher in those who perceived their health status as "very bad-bad." The poor SQ was 1.45 (95% CI [1.05–2.00]; $p = 0.02$) times higher in those who perceived their family's economic status as "moderate" compared to those who perceived their economic status as "very good-good." In addition, poor SQ was 1.83 (95% CI [1.22–2.74];

**Table 5 A logistic regression analysis among variables influencing poor sleep quality.**

| | | Crude OR (95% CI) | Adjusted OR ** (95% CI) | p |
|---|---|---|---|---|
| Age (year) | ≤15* | 1.00 | **1.00** | |
| | ≥16 | 1.36 (1.04–1.77) | **1.39 (1.05–1.84)** | **0.021** |
| Gender | Male* | 1.00 | 1.00 | |
| | Female | 1.59 (1.22–2.09) | **1.50 (1.13–2.00)** | **0.005** |
| Paternal education level | University* | 1.00 | 1.00 | |
| | ≤High school | 1.36 (1.04–1.78) | 0.82 (0.58–1.16) | 0.280 |
| Maternal education level | University* | 1.00 | 1.00 | |
| | ≤High school | 1.11 (0.85–1.46) | 1.29 (0.91–1.81) | 0.145 |
| Perceived health status | Very good-good* | 1.00 | 1.00 | |
| | Moderate | 1.81 (1.20–2.75) | **1.74 (1.12–2.69)** | **0.013** |
| | Bad-very bad | 6.03 (1.97–25.47) | **6.39 (1.83–22.30)** | **0.004** |
| Perceived economic status of the family | Very good-good* | 1.00 | 1.00 | |
| | Moderate | 1.55 (1.16–2.07) | **1.45 (1.05–2.00)** | **0.021** |
| | Bad-very bad | 1.63 (0.61–4.77) | 0.90 (0.29–2.71) | 0.854 |
| Internet addiction | Normal* | 1.00 | 1.00 | |
| | Problematic | 2.39 (1.66–3.46) | **1.83 (1.22–2.74)** | **0.003** |
| | Pathological | 3.67 (2.08–6.79) | **1.99 (1.23–3.87)** | **0.043** |
| SAS total score | *Every 1-point increment* | – | **1.01 (1.00–1.02)** | **<0.001** |

$R^2 = 0.150$; Constant B = −1.527; S.E. = 0.266; Exp(B) = 0.217 ($p < 0.0001$)

Notes:
The students' age, gender, parents' education, perceived health status, perceived economic status of family, IA level, and total score of SAS were evaluated together in binary logistic regression (enter method).
*Reference value.
**It has been adjusted according to the variables included in the model.
Significant $p$ values were shown in bold.

$p < 0.01$) times higher in those with problematic internet use than those with normal internet use scores, and 1.99 (95% CI [1.23–3.87]; $p = 0.04$) times higher in those with pathological internet use. A one-point increase in the SAS total score increased poor SQ 1.01 (95% CI [1.01–1.02]; $p < 0.01$) times. The education level of the students' parents did not affect poor SQ (Table 5).

## DISCUSSION

Sleep is an indispensable need for Turkish adolescents to achieve physical growth and improve academic performance (*Brown et al., 2018*). Adolescents must sleep sufficiently to achieve their developmental functions (*Tarokh, Saletin & Carskadon, 2016*). In this study, we investigated the factors affecting poor SQ in adolescents and the relationship between IA and SA with the poor SQ. We found that IA, SA, the students' age group, gender, perception of health status, and perception of their family's economic status affect poor SQ.

Sleep is important for the cardiovascular and hormonal systems of adolescents whose growth and development are not yet completed (*Brown et al., 2018*). Therefore, changes in SQ may affect morbidity and academic success (*Brown et al., 2018*). SQ of approximately two-thirds (58.7%) of the students participating in our study was poor. In studies conducted with Turkish adolescents using PSQI, the frequency of poor SQ was similar to the result of our study (54.7%, 58.6%) (*Akçay & Akçay, 2018*; *Koças & Şaşmaz, 2018*). In another study conducted with medical students, the frequency of poor SQ was 62.5% (*Demirci et al., 2015*). In studies conducted with adolescents in different countries, the frequency of poor SQ varies between 9.8% and 60.4% (*de Lima & Silva, 2018*; *Huang et al., 2020*; *Kurugodiyavar et al., 2018*; *Lin et al., 2019*; *Salama, 2020*). In the literature, it is stated that demographic characteristics such as age, gender, and sociocultural factors are also effective in adolescents' sleep (*Huang et al., 2020*; *Koças & Şaşmaz, 2018*; *Wang et al., 2019*; *Özcan, 2020*). In this study, we found that adolescents aged 16 and over had 1.4 times poorer SQ than those under 16 years of age. With this result, the third hypothesis of our research was confirmed. Studies conducted with adolescents indicate that the frequency of poor SQ increases with age (*Demirci et al., 2015*; *Wang et al., 2019*). There are also studies showing that there is no relationship between the age of adolescents and poor SQ (*Huang et al., 2020*; *Kurugodiyavar et al., 2018*; *Lin et al., 2019*). In the adolescent age group, sleep is a physiological need that may affect thought and attention-seeking activities (*Brown et al., 2018*; *de Lima & Silva, 2018*). Therefore, determining the factors affecting poor SQ and raising awareness in adolescence are important to create positive behavioral changes.

As an interesting finding in our study, poor SQ was 1.5 times higher in female students compared to male students participating in our study. With this result, the third hypothesis of our research was confirmed. Gender differences may affect the sleep patterns of adolescents. But, there is no consensus in the literature regarding the relationship between poor SQ and gender in adolescents. In several studies, poor SQ is reported to be more common in male adolescents (*Demirci et al., 2015*; *Huang et al., 2020*; *Kurugodiyavar et al., 2018*). In others, adolescent girls are more prone to SQ problems than adolescent boys (*Demirci, Akgönül & Akpinar, 2015*; *Koças & Şaşmaz, 2018*; *Wang et al., 2019*). Similar to the result of our study, poor SQ was found 1.7 times higher in female students in a study conducted on adolescents in Brazil (*de Lima & Silva, 2018*). In addition to hormonal changes observed in adolescent girls, changes in health-related behaviors and electronic device use also negatively affect SQ (*Bruni et al., 2015*; *Lee, Kim & Choi, 2017*; *Lin et al., 2019*; *Wang et al., 2019*). Additionally, irregular menstrual cycles due to hormonal changes and waking up habits at night due to menstrual pain may adversely affect the SQ

of female adolescents (*Kazama, Maruyama & Nakamura, 2015*; *Nam, Han & Lee, 2017*). Although it is not in the scope of this study, health-related behaviors, self-actualization, interpersonal support, and stress management behavior observed in adolescent girls may also affect SQ.

Poor SQ may cause mood, behavior, memory, and attention problems in adolescents. The physiological and psychological effects of diseases may impair the duration and quality of sleep. Diseases that cause physical discomfort or problems such as anxiety and depression may be related to sleep problems. In studies conducted with adolescents, those who had smoking habits (*Lin et al., 2019*), who were diagnosed with depression (*Huang et al., 2020*; *Onat et al., 2019*), who had physical and mental health problems (*Huang et al., 2020*), and who cannot cope with stress (*Wang et al., 2019*) were found to have poorer SQ. In our study, we identified that students who were not satisfied with their health had poor SQ. With this result, the third hypothesis of our research was confirmed. Identifying and treating the underlying causes of problems affecting general health in adolescence may help improve SQ.

Another finding of our study, we found a relationship between the perception of the family's economic status and the SQ of Turkish adolescents. Poor SQ was common in those who perceived their economic situation as bad. With this result, the third hypothesis of our research was confirmed. However, our results could not confirm one of the third hypotheses of our study, that 'parental education level negatively affect SQ'. It is reported that low socioeconomic level is significantly associated with short sleep (*Knutson, 2013*). However, there are also studies in the literature that do not find a relationship between adolescents' perception of the family's economic situation and their family's education level and poor sleep quality (*Huang et al., 2020*; *Onat et al., 2019*; *Demirci et al., 2015*). The socioeconomic and cultural structure of the family may affect the physical and mental health of adolescents. This situation may cause a deterioration in SQ among adolescents.

In recent years, computer and internet usage has become an indispensable part of life. Today, adolescents spend a lot of time on the internet (*Bilgrami et al., 2017*). The internet is considered a technological miracle that supports young people's access to information, research, problem-solving, creativity, and critical thinking (*Bilgrami et al., 2017*). However, excessive internet use brings many health problems. There are many studies examining the relationship between IA and SQ in the literature: In a study conducted on college students in Taiwan, the SQ of students with IA was 1.05 times worse than students without IA (*Lin et al., 2019*). A study conducted on medical students revealed that students with sleep disorders had higher IA scale scores (*Demirci et al., 2015*). In studies conducted in different areas of the world, IA has been shown to cause poor SQ and sleep problems in adolescents (*Akçay & Akçay, 2018*; *Bruni et al., 2015*; *Ekinci et al., 2014*; *Koças & Şaşmaz, 2018*; *Onat et al., 2019*; *Salama, 2020*). Similarly, in our study, poor SQ was 1.8 times higher in students with problematic internet use and 2.0 times higher in students with pathological internet use compared to students who were not internet addicts. With this result, the first hypothesis of our research was confirmed. In a study, conducted on a sample similar to our study, results were similar to our findings (*Koças & Şaşmaz, 2018*). Accordingly, IA is an important risk factor for poor SQ. Light and sound coming from devices such as

computers, tablets, and smartphones may cause poor SQ by disrupting the individual's sleep rhythm, delaying the transition to sleep, shortening the time remaining for sleep, and causing breaks during sleep (*Koças & Şaşmaz, 2018*). IA and insufficient sleep may cause unwanted consequences in daily life and school life (*Salama, 2020*; *Wang et al., 2019*). In addition, there may be multiple and complex interactions between IA and mental problems such as anxiety, depression, and stress (*Choi et al., 2015*; *Lachmann et al., 2017*; *Salama, 2020*). The prevalence of IA was high in the adolescents participating in our study. It may be important to investigate this issue, as IA may cause mental problems as well as the deterioration of SQ.

An important finding in our study was the relationship between SA and poor SQ. Adolescents mostly use smartphones for social networking, messaging, and internet access (*Demirci, Akgönül & Akpinar, 2015*; *Lee, Kim & Choi, 2017*). Therefore, in the literature, SA is more common in adolescents with IA (*Choi et al., 2015*). In our study, students with poor SQ had higher SAS total and subcomponent scores (daily-life disturbance, positive anticipation, withdrawal, cyberspace-oriented relationship, overuse, and tolerance). We found a positive correlation between the SAS total score and the PSQI subcomponents (subjective SQ, sleep latency, sleep duration, sleep disturbances, use of sleep medication, and daytime dysfunction). With this result, the second hypothesis of our research was confirmed. SA and the habit of using smartphones negatively affect SQ and cause poor SQ in adolescents (*Demirci, Akgönül & Akpinar, 2015*; *Huang et al., 2020*; *Kang et al., 2020*; *Kurugodiyavar et al., 2018*; *Wang et al., 2019*; *Xie, Dong & Wang, 2018*). In a one-year follow-up study of Chinese adolescents, no relationship was found between SA and anxiety and depression (*Kang et al., 2020*). However, in another study conducted in China, a significant relationship was shown between problematic smartphone use and eye problems, immune system disorders, and fatigue symptoms (*Xie, Dong & Wang, 2018*). In other studies, SA in adolescents is shown to be an important risk factor for mental illnesses such as stress, anxiety, and depression (*Demirci, Akgönül & Akpinar, 2015*; *Panova & Carbonell, 2018*). The prevalence of SA was high in the adolescents participating in our study. Since SA may cause various health problems as well as the deterioration of SQ, studies investigating the relationship of SA with various diseases may provide significant benefits. Considering that SA is a public health problem, a program that will raise awareness in students and their families with a multidisciplinary approach should be created. Proper use of technology, technology addiction, time management, and the importance of healthy living habits may be added to the curriculum. In order to reduce the time they spend on smartphones and internet, opportunities to engage in activities related to culture, sport, and art should be promoted. Initiatives to be implemented to prevent SA and IA in adolescents may also improve SQ. Additionally, counseling which aims at the behavioral change to increase the SQ of students is strongly recommended.

We agree with the literature that excessive use of the Internet and mobile phones negatively affects sleep (*Bruni et al., 2015*; *Huang et al., 2020*; *Kang et al., 2020*; *de Lima & Silva, 2018*; *Lin et al., 2019*; *Wang et al., 2021*; *Xie, Dong & Wang, 2018*). However, it is an interesting result that there is no relationship between IA and SA and habitual sleep efficiency, which is one of the subcomponents of SQ. In the literature, there are studies showing that the

 

habitual sleep time is longer in women than in men (*Knutson, 2013*), there is no difference between the obese and control groups in terms of habitual sleep efficiency (*Eliacik et al., 2016*), and habitual sleep efficiency is higher in depressed adolescents (*Onat et al., 2019*). No explanatory information was found in the literature on this subject. In order to determine the reasons for this relationship, it would be appropriate to conduct follow-up studies in which sleep quality in adolescents is determined by measurement.

This study had some limitations. First, all data were obtained from self-reported questionnaires. The recall factor may have played a role. However, this limitation cannot be interpreted as a random error and cannot be considered as bias. Because the same factor is effective for those with good sleep quality. Second, due to the limitations of the cross-sectional study, we could only determine the relationship between individual variables, but not the exact cause-effect relationships. Therefore, a follow-up study may be conducted to reveal the exact relationship.

## CONCLUSIONS

In our study, IA and SA were associated with the poor SQ in Turkish adolescents. Age (≥16 years), gender (female), bad health status, and moderate economic status of the family were other factors that were associated with the poor SQ. Identifying students in the adolescent age group with poor SQ should be the main step in addressing the issue. Conducting interventional studies examining the effect of cultural environment and behavioral treatments on SQ, IA and SA may be important in understanding and solving the problem.

## ACKNOWLEDGEMENTS

The authors would like to thank students and their families. The authors did not receive support from any organization for the submitted work.

### Funding

The authors received no funding for this work. The funders had no role in study design, data collection and analysis, decision to publish, or preparation of the manuscript.

### Competing Interests

The authors declare that they have no competing interests.

### Author Contributions

- Ayla Acikgoz conceived and designed the experiments, performed the experiments, analyzed the data, prepared figures and/or tables, authored or reviewed drafts of the paper, and approved the final draft.
- Burcu Acikgoz conceived and designed the experiments, performed the experiments, authored or reviewed drafts of the paper, and approved the final draft.
- Osman Acikgoz conceived and designed the experiments, performed the experiments, authored or reviewed drafts of the paper, and approved the final draft.

# PeerJ

## Human Ethics

The following information was supplied relating to ethical approvals (*i.e.*, approving body and any reference numbers):

Ethical approval was obtained from the Non-invasive Research Ethics Committee of Dokuz Eylul University, Izmir, Turkey (Decision No: 2018/01-39).

## Data Availability

The raw data is available in the Supplemental File.

## Supplemental Information

Supplemental information for this article can be found online at http://dx.doi.org/10.7717/peerj.12876#supplemental-information.

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
