# Peer review of "The effect of internet addiction and smartphone addiction on sleep quality among Turkish adolescents"

_PeerJ, doi:10.7717/peerj.12876_

## Round 0.1 · original submission · Major Revisions

Your manuscript has been reviewed and assessed by two reviewers, and both of them agree with the fact that there are still a few points that need to be addressed. The comments of the reviewers are included at the bottom of this letter. Reviewers indicated that methods, results, and discussion sections should be improved. Reviewers also recommended extensive English editing. We would be glad to consider a substantial revision of your work, where the reviewer’s comments will be carefully addressed one by one. In addition to these; please see my comments below:

- Hypotheses were not clear, Why this study was very important to address? What was the gap in knowledge?

- It should be explained how three schools where the research was applied were selected. Which sampling method was used to determine the sample?

- The methods section needs more elaboration, in particular, data analysis. For example, did you perform a step-wise/forward/backward logistic regression? How did you check the normality assumption of the data? I checked the normality assumption of scores for SAS subscales and total in sleep quality groups and found that data were not normally distributed. Therefore student t-test should not be used to compare the total scores of two independent groups. Please check your methods section with an expert biostatistician.

- Please provide Cronbach alpha values of each scale in your study.

- Line 141: Add “IBM” before “Statistical 142 Package of the Social Science for Windows 24.0”.

- What was the power of the study? Which statistical package or tool was used to calculate sample size? Please provide the name of the package or the tool.

Reviewer 1 ·

Basic reporting

In general, the hypothesis of the research is worths studying for adolescent health.

Experimental design

The design is well organised.

Validity of the findings

The findings are clear and valuable.

Additional comments

1. The English language needs a reduction in some sentences, such as in the introduction section because internet access becoming easier; the risk of excessive internet use has increased.’
2. Methods section was well organized. However, it is not clear what kind of questions were filled to determine the perceived socioeconomic status.
3. In statistical methods there is a need to write the form of logistic regression analysis.
4. In the results section, the first sentence needs correction. In the beginning of the paragraph, we cannot use abbreviations. ‘SQ of 58.7% of the students was poor, 41.3% of them were good.’. And in the same paragraph, we cannot give the result of the p-value as p<0.05. And in the continuing results avoid using p-value as p<0.05. If you would like to emphasize that all results are statistically significant, just write it in words.
5. In the results section the sentence ‘Factors affecting the poor SQ of the students were analyzed using the logistic regression model.’ Needs correction. Try to change it to a result sentence.
6. Tables are well organized, except writing the method of logistic regression analysis.
7. Studies that investigate IA and/or sleep quality in adolescents are limited in Turkey. I recommend utilizing the paper ‘Internet addiction, sleep and health-related life quality among obese individuals: a comparison study of the growing problems in adolescent health. Eliacik et al. Eating and Weight Disorders - Studies on Anorexia, Bulimia and Obesity volume 21, pages709–717 (2016)
8. In the discussion section avoid comparing your results with the literature. In order to do that try to explain your results.

·

Basic reporting

Use of English language

To ensure that the international audience can clearly understand your text, I suggest that you consider improving the use of English language in the following aspects:

1. Avoid the repetition of terms/ideas in the same sentence or in close sentences. Some examples include lines 41-42, 42-43, 88-90, 98-99, 121-122, 130-132, 189-190, 201, 225-228.

2. In certain cases, you can put more appropriate terms for what you intend to say. Some examples include lines 78 (“[Many] studies…”), 98 (“declared by [each] student”), 212 (“In [other] studies…”), 240 (“There are [many] studies examining the relationship…”).

3. Improve the use of verb tenses. Some examples include lines 35 (“Data [was] collected…”) and 60 (“Internet access [is] becoming”).

I suggest you have a colleague who is proficient in English and familiar with the subject matter review your manuscript, or contact a professional editing service.

Introduction and background

1. I emphasize the need that the contents cited in the Discussion (and debated with the results obtained), should also be mentioned in the Introduction. For example, in lines 188-189, the effect of sleep on the cardiovascular and hormonal systems appears for the first time, although it should also be mentioned in the Introduction.

2. Be aware of situations where you give scientifically unproven opinions. Some places that need references are lines 78, 183-184, 218-219, 243.

3. The investigation hypotheses were not clear to me. What are your study hypotheses? What do you expect to find in this study? This should be referred in the Introduction and based on scientific literature.

4. The article should include sufficient introduction to demonstrate how the work fits into the broader field of knowledge. In this sense, I consider that it is important to designate what smartphone addiction is, to refer studies that address the factors that influence SQ (gender, age, parental education, etc.) and to illustrate what are the conclusions of the studies about IA and SA in adolescents in Turkey.

5. The elements of connection between paragraphs and phrases are important for the reader to understand a common thread of what is being said. I suggest you think about this in lines such as 65-66, 83-84, 234-235.

6. In Abstract’s background (lines 32-34), I suggest that you try to make the reader understand why it is important to study the effect of internet addiction and smartphone addiction on the sleep quality of Turkish adolescents. Still in the Abstract, in conclusions, it would be important to highlight the main variables (i.e., internet addiction and smartphone addiction) instead of place them mixed with age, gender, perceived health status, and perceived economic status of the family.

7. For a general understanding of the text that follows, I suggest you put brief introductory sentences on lines 57 and 153.

8. Sometimes I see a need for more concrete data. For example, in line 68, what are the numerous benefits of smartphones?

Raw data supplied

The order of the tables is well placed, as is the position of the its titles. I suggest, however, the consideration of the following changes:

1. Improve the presentation of the results, because data mentioned in Results’ section does not correspond to what is presented on tables. Also, it would be important to complete the tables with more relevant information such as the value of test statistics.

2. Improve table 4, where correlations are referred. To have a better understanding of the data you can put, in "Notes:", asterisks that indicate when the relationship between variables was significant (i.e., *p<0.05) or very significant (i.e., **p<0.001). In this sense, I suggest that you remove from the table the lines referring to p. Please write in this table that the top horizontal line refers to the Pittsburgh Sleep Quality Index. Still, I emphasize that it is normal to place, both horizontally and vertically, all the variables used. I suggest that you look at a table with correlations in a PeerJ article.

3. Review details that increase clarity and professionalism in scientific writing. For example, in table 3, remove the colons in “Good” and “Poor”, since it has parentheses after; in all tables, put p in italics; in the section of “Results”, describe the content of each table in more detail; write “Notes:”, before referring them below the tables; remove words that are unnecessary, such as "internet use", in table 2.

Experimental design

This article corresponds to the aims and scopes of the journal. It’s ethical procedures also seems adequate to me. Nevertheless, I suggest that you consider the following aspects:

1. To better understand the importance of the present study, it would be important to reflect on the following question: Why is it important to study the effect of IA and SA on SQ in Turkish adolescents? I believe that the answer to this question can improve your research question. I also think that your research aims can be clearer and more congruent. Note that the Introduction prescribes that it is intended to study the effects of IA and SA on SQ in Turkish students, but it is stated in the Discussion that you also studied other factors that affect SQ.

2. To better replicate your results, it would be important to mention whether the three high schools were public or private, as well as when the questionnaires were filled out by the participants (during or after school?). You should consider noting and justifying the creation of the two groups of adolescents (i.e., aged 15 or under and aged 16 or over).

3. In the section Materials & Methods, you described each instrument, but you could also describe in more detail what each subscale means.

4. In the last sentence of the section Introduction, you can write something like "This study aimed to address this gap.". In this way, the research question is better described and the repetition of what was mentioned in the previous sentence is avoided.

Validity of the findings

1. Correlation differs from a cause-and-effect relationship. That's why you can't put "may cause" (lines 224-225) if you then address an association between variables.

2. In the Conclusions, to refer “older students” is ambiguous. I suggest that you affirm the age group. Still in Conclusions, it would be important to better clarify the issue that you want to address.

Additional comments

1. Pay attention to inconsistencies. If you don't put an acronym on the other instruments, don't put it on Smartphone Addiction Scale (SAS) – line 36.

2. In the Discussion it may be important to recall the subscales of each scale, so that the readers can remember them easier.

I wish you a good work.

---

## Round 0.2 · Major Revisions

The manuscript has been assessed by two reviewers, and one of them agrees with the fact that there are still a few points that need to be addressed. We would be glad to consider a substantial revision of your work, where the reviewer’s comments will be carefully addressed one by one.

Reviewer 1 ·

Basic reporting

Clear and unambiguous, professional English is used throughout.

Experimental design

Research question well defined, relevant & meaningful. It is stated how research fills an identified knowledge gap.

Validity of the findings

All underlying data have been provided; they are robust, statistically sound, & controlled. Conclusions are well stated, linked to original research question & limited to supporting results.

·

Basic reporting

In the abstract, I suggest the following, to becomes clearer:
- In the introduction, start the sentence with "Compared to participants with normal internet addiction scores, sleep quality was..."(lines 48-51).
- In the results part, it is not necessary to write "total subscale scores of" and "the total score of.". Just put "...were positively correlated with smartphone addiction and internet addiction” (line 52-54).
- When you’re presenting the conclusions, write "older adolescents" instead of “older age” (line 56)
To demonstrate a professional writing, avoid writing something like "15-24". For example, in line 68, put "aged between 15 and 24 years old".

I suggest you improve the definition of internet addiction. Problems in life and in psychological domains are part of the definition (line 73-75). I consider that you can also develop a little more about the internet addiction, smartphone addiction and sleep quality, instead of just describing the components of them. This will support the importance of your study.

I noticed that some connecting words are missing. I highlight the following examples: “associated [with] poor sleep quality” (line 55), “education, [and] entertainment”, “daily life of [its] users.” (line 84).
“Being an environment” is not the right term. Maybe you wanted to say, “being a means for socialization” (line 80).

Between lines 86 and 87, it is necessary to place some connecting element. It needs to be clear why are you starting to talk about the sleep quality. What is the relationship with what you talked about above? It’s also necessary to place a connecting element between lines 95 and 96.

I suggest that you build better the sentence of line 93. It must become explicit why you chose to talk about (and study) sociodemographic determinants and not others. And what is the conclusion of the author about the sociodemographic determinants of SQ?

I suggest that you better clarify the gap in the scientific literature. To do that you can explain, in line 110, the problem is not just the few studies, but also the fact that many of them have limitations. You can also clarify why it’s important to study this theme in Turkish adolescents (line 120; you can highlight the fact that this population has high rates of internet and smartphone use; you can also mention how this study can benefit them).

I suggest that you better clarify the relationship between your hypotheses and the scientific literature. Why do you want to study adolescent’s sociodemographic and family characteristics?

Discussion is not very fluid. I suggest that you invest in connecting elements between paragraphs.

In the discussion, in lines 272-273 it remains to put a reference.

In the discussion, I suggest that you write “we identified”, instead of determined (line 303). This will improve your professional writing.

Experimental design

When you refer your instruments, don’t forget to cite their authors (e.g., lines 142-144, 160,169, 191).

In line 146 it’s more correct to put “Informed consent”.

When you say that it was evaluated socioeconomic and health-related characteristics, it is important to mention that you evaluated perceptions. Also, don’t forget to mention how do you evaluated perceived health status and, to achieve a professional writing, improve line 150 by referencing the Likert scale used.

Lines 151 and 152 mention an important topic to be addressed in the introduction. Also, to better understand your division into two groups of adolescents, you should mention in the introduction between what ages is adolescence considered.

Don't forget to put the Likert scale of the Pittsburgh Sleep Quality Index (lines 190-203).
The presentation of results needs more detail. For example, when you say, "some subcomponents", you should mention which ones (line 240). Also, it is important to note that, as shown in Table 1, gender, age, paternal education level, perceived health status and perceived economic status of family are significantly associated with sleep quality (See, for example, the following PeerJ article, which used chi-square test: Saraswathi I, Saikarthik J, Senthil Kumar K, Madhan Srinivasan K, Ardhanaari M, Gunapriya R. 2020. Impact of COVID-19 outbreak on the mental health status of undergraduate medical students in a COVID-19 treating medical college: a prospective longitudinal study. PeerJ 8:e10164 DOI 10.7717/peerj.10164).

In the perceived economic situation description, you indicate that there are five possible answers, but in the presentation of the results (including the tables), there were only three. I suggest that you clarify why you made the decision to talk about "very bad-bad" and "very good -good".

Validity of the findings

Don’t forget to mention, in the abstract's results, that one of the subcomponents of sleep quality (habitual sleep efficiency) was not positively related to smartphone addiction and internet addiction (lines 52-54). Also, in the abstract, clarify in the conclusions that poor health and economic status of the family are perceptions (line 56-57).

In the Discussion and in the Conclusions, it’s not clear that you studied Turkish adolescents.
I suggest that you mention your hypotheses in the Discussion. Which were confirmed and which weren’t?

In line 266, it should contain the correct percentage.

In the paragraph corresponding to lines 306 to 310, what does the scientific literature say about this?
In the Discussion, you should discuss the unexpected results. For example, why you didn't find a significant relationship between smartphone and internet addiction with habitual sleep efficiency? What is shown in the scientific literature?

I suggest that you think about a solution for your first limitation (line 361-362).

In the discussion, I notice that some cited results are not in the literature review. Don’t forget that the content of the articles presented in the discussion should be mentioned in the introduction. In the discussion part, you relate your results with the literature review that you wrote in the introduction.

---

## Round 0.3 · Minor Revisions

Thank you very much for the submission of a revised version of your paper. I have gone through the revised, track-changes manuscript and rebuttal letter, and see that the authors addressed the reviewers' concerns and substantially improved the content of the manuscript. So, based on my own assessment as an academic editor, the manuscript is almost ready to be accepted for publication.

Please see my comments below:

-Please provide p-values at the end of the sentence “As shown in Table 1, gender, age, paternal education level, perceived health status and perceived economic status of the family are significantly associated with sleep quality.” in the first paragraph of the results section

-Share p-values and correlation coefficients at the end of the sentences in the fourth paragraph of the results section, for example: “(r=0.789; p=0.03)”

-Give p-values at the end of the sentences in the last paragraph of the results section after CI values, for example: “(95% CI: 1.13-2.00; p=0.01)”

---

## Round 0.4 · accepted · Accept

Thanks for revising your manuscript based on the concerns. I now believe that your manuscript is suitable for publication. Congratulations! I look forward to seeing this work in print. Thanks again for choosing PeerJ to publish.